# ReX: A Framework for Incorporating Temporal Information in Model-Agnostic Local Explanation Techniques

## Abstract

Advanced machine learning models that can handle inputs of variable lengths are powerful, but often hard to interpret. The lack of transparency hinders their adoption in many domains. Explanation techniques are essential for improving transparency. However, existing model-agnostic general explanation techniques do not consider the variable lengths of input data points, which limits their effectiveness. To address this limitation, we propose ReX, a general framework for adapting various explanation techniques to models that process variable-length inputs, expanding explanation coverage to data points of different lengths. Our approach adds temporal information to the explanations generated by existing techniques without altering their core algorithms. We instantiate our approach on three popular explanation techniques: Anchors, LIME, and Kernel SHAP. To evaluate the effectiveness of ReX, we apply our approach to five models in two different tasks. Our evaluation results demonstrate that our approach significantly improves the fidelity and understandability of explanations.

## 1 Introduction

As more critical applications employ machine learning systems, how to explain the rationales behind their results has emerged as an important problem. Such explanations allow end users to 1) judge whether the results are trustworthy (Ribeiro et al., 2016; Doshi-Velez et al., 2017) and 2) understand knowledge embedded in the systems so they can use the knowledge to manipulate future events (Poyiadzi et al., 2020; Prosperi et al., 2020; Zhang et al., 2018). This paper focuses on the problem of explaining deep learning systems that are widely applied to processing sequential data of different lengths, such as Recurrent Neural Networks (RNNs), Transformers(Vaswani et al., 2017; Wolf et al., 2020), and others.

Explanation techniques can be classified as *global* or *local* (Molnar, 2020). The former explains how the model behaves on all inputs, while the latter explains how the model behaves on a particular set of inputs (typically ones that are similar to a given input). Most of the existing explanation techniques for variable-length models are global. Take techniques for RNNs as an example: they employ deterministic finite automaton (DFAs) as global surrogates of target RNNs (Omlin & Giles, 1996; Jacobsson, 2005; Wang et al., 2018; Weiss et al., 2018; Dong et al., 2020). However, due to the complex nature of any practical problem domain, these techniques can produce very large DFAs. These explanations are hard for a human to digest. Moreover, they take a long time to generate, making these techniques hard to scale. Therefore, these techniques are often limited to toy networks such as ones that learn regular expressions. Even for these simple domains, the explanations can still be complex as RNNs often fail to internalize the perfect regular expressions and contain noise.

Due to the complexity of the problem domain and the networks, we turn our attentions to local methods (Ribeiro et al., 2016; 2018; Zhang et al., 2018; Arras et al., 2017; Wachter et al., 2017; Lundberg & Lee, 2017). Local methods produce more tractable and understandable explanations at the cost of covering much fewer inputs. Such methods provide explanations for individual inputs and have a wide range of applications. While there is a rich body of techniques in this category, to our knowledge, there are few that are specialized to capture the temporal information of inputs, which

Table 1: Example explanations generated in Anchors (a) without REX and (b) with REX.

| Input Sentence | | I. He never fails in any exam. | II. He never attends any lecture, so he fails in any exam. |
|---|---|---|---|
| Network output | | Positive | Negative |
| Explanations: | (a) | {never, fails} | {never, fails} |
| | (b) | {never, fails} $\wedge \text{Pos}_{\text{fails}} - \text{Pos}_{\text{never}} = 1$ | {never, fails} $\wedge \text{Pos}_{\text{fails}} - \text{Pos}_{\text{never}} \geq 2$ |

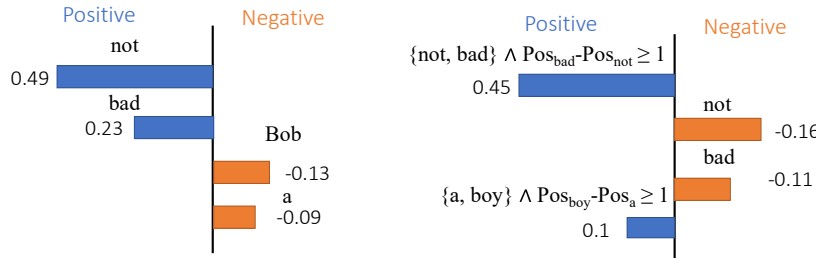

**Input sentence** :  Bob is not a bad boy.    **Network output**:  Positive
**Explanations**:

Figure 1: An example explanation generated in LIME without REX (left) and with REX (right).

plays an important role in models like RNNs and transformers. Without incorporating temporal information, the explanations lack fidelity and understandability.

Consider Anchors (Ribeiro et al., 2018), a popular local technique that generates sufficient conditions to explain specific outputs. Table 1 shows Anchors explanations of a sentiment analysis RNN on two sentences. For sentence I, Anchors generates the following explanation: *Sentence I is positive because it contains both "never" and "fails"*. For sentence II, Anchors produces the same explanation as for Sentence I, but the sentence is negative. The key difference is in Sentence II, "never" and "fails" do not form a phrase. Anchors fails to capture the difference, which confuses users.

We see similar issues with other existing local explanation techniques. For example, attribution-based techniques like LIME (Ribeiro et al., 2016) use linear models as local surrogates. Figure 1 shows LIME explanations of a sentiment analysis transformer on a sentence. LIME assigns a high positive score to "not" and "bad". Intuitively, it means that either "not" or "bad" can make the sentence positive. But this will lead to some ridiculous results. The user will predict "Bob is a bad boy" as a positive sentence since the word "bad" should have a strong positive effect. However, this is not the case. "Not bad" together is a positive phrase, while "not" or "bad" alone is a negative word.

To address this issue, we propose REX, a general framework for extending various local *model-agnostic* explanation techniques with temporal information. In particular, REX adds temporal information in the form of $Pos_f \; op \; d$ and $Pos_f - Pos_g \; op \; d$ ($op$ can be $=, >, <, \geq$, or $\leq$ ), where $Pos_f$ means the position of feature $f$. The way to present temporal information depends on the specific explanation technique. Consider the sentences in Table 1 again. After REX augments Anchors, the explanation to Sentence I becomes *the sentence is positive because it contains both "never" and "fails", and "never" is right before "fails"*. For sentence II, the explanation becomes *the sentence is negative because it contains "never" and "fails", and "never" is not right before "fails"*.

Now, consider applying LIME with REX to the sentence in Figure 1. The fact that *"not" is before "bad"* gets the highest positive score, while "not" and "bad" both get negative scores respectively. This new information 1) associates the two words, and 2) captures that "not" comes before "bad".

Figure 2 shows another example when explaining an anomaly detection RNN. Anchors generates an explanation: *the anomaly is detected because of the presence of several separated data points.* After REX augments Anchors, the explanation becomes *the anomaly is detected because of the presence of data points 413 and 425 with at least 3 points between them.* Compared to the original explanation, the new explanation is more general and more understandable to end users.

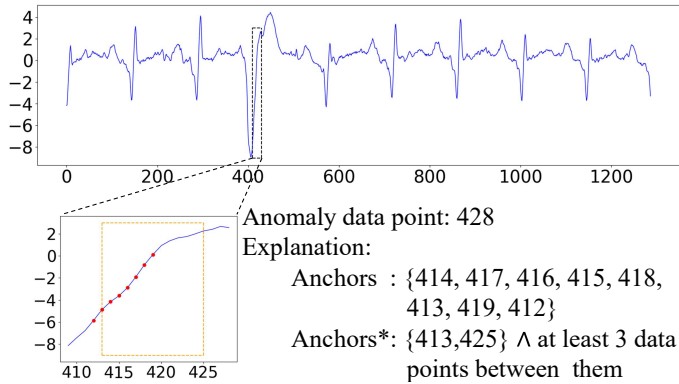

Figure 2: Explaining an anomaly detection RNN with Anchors and Anchors*(Anchors with REX).

The examples show that temporal information improves the fidelity of explanations and makes them easier to understand. How to add such information to various local model-agnostic explanation techniques? We notice that these techniques treat target models as black boxes and generate surrogates by learning from input-output pairs. In particular, we make two key observations: 1) these techniques use a perturbation model to generate inputs that are similar to the original input so it captures the local behavior of the model via these inputs; 2) these techniques generate explanations that are described in features. Based on 1), in order to capture temporal information which the model internalizes, we can modify the perturbation model to generate inputs whose lengths and feature value orders vary. Based on 2), we can treat temporal information such as $Pos_f - Pos_g \geq d$ as a predicate used to form the explanation. Although this feature will not affect the result of the original network, they reflect temporal information that is internalized by the network. More importantly, now the explanation techniques can generate explanations that utilize them without changing their core algorithms.

To evaluate the effectiveness of REX, we have applied it to LIME, Anchors, and kernel SHAP (Lundberg & Lee, 2017) and then use their augmented versions to explain a sentiment analysis LSTM, several sentiment analysis transformer models (BERT (Devlin et al., 2018),T5 (Raffel et al., 2020), and GPT 2.0 (Radford et al., 2019)), and an anomaly detection RNN. Our results show that REX improves existing techniques in both the fidelity and understandability of generated explanations. On average, REX helps improve the fidelity of Anchors, LIME, and Kernel SHAP explanations by 98.2%, 28.6%, and 23.1% respectively. Moreover, a user study shows that augmented explanations make it easier for human users to predict the behaviors of the models.

In summary, we have made the following contributions:

- We have proposed to incorporate temporal information in the form of $Pos_f \; op \; d$ and $Pos_f - Pos_g \; op \; d$ in local explanations to machine learning models that can capture temporal information in the inputs, which makes these explanations more faithful and easier to understand.
- We have proposed a general framework REX to automatically incorporate the above information in popular local explanation techniques.
- We have demonstrated the effectiveness of REX by applying it to LIME, Anchors, and kernel SHAP, and evaluating the augmented versions on representative models.

## 2 PRELIMINARIES

In this section, we describe the necessary background to introduce our approach. Without loss of generality, we assume the target model is a black-box function from a sequence of real numbers to a real value, $f : \mathbb{R}^* \to \mathbb{R}$, where $\mathbb{R}^* = \bigcup_{T \in \mathbb{N}} \mathbb{R}^T$. We limit our discussion to classifiers and regressors.

Given an input $x \in \mathbb{R}^*$, its corresponding output $o \in \mathbb{R}, o := f(x)$, a local explanation is $g(x, f)$. It reflects why the model produces output $o$ given input $x$ and the target model's behavior around input $x$. Typically, a local explanation is a function of input $x$ and model $f$. Specifically, $g(x, f)$ predict the output of model f on another input $x'$ based on the evaluation result of an expression formed with **feature predicates** $\{p_j\}_{j=1,2...,|x|}$. Formally, we define $p_j := f_j \; op \; c$, where $f_j$ represents

the $j$th feature of input $x$, $op$ is a binary operator (e.g., $=, \geq, \leq$), and $c$ is a constant. The feature predicate $p_j$ compares the $j$th feature of input $x$ to the constant. Based on the specific form of the expression, a feature predicate can be interpreted as a Boolean variable or a $\{0, 1\}$ integer. We call the set of feature predicates the **vocabulary** of the local explanation. For example, in Anchors, $g(x, f)$ is a conjunction $\bigwedge_{j \in \{1,2,...,|x|\}} p_j$ which must evaluate to true on $x$; in LIME and kernel SHAP, $g(x, f)$ is a linear expression in the form of $\Sigma_{j=1}^{|x|} \omega_j p_j$; in a counterfactual explanation (Zhang et al., 2018; Wachter et al., 2017; Dandl et al., 2020), $g(x, f)$ is a conjunction $\bigwedge_{j \in \Delta} p_j \wedge \bigwedge_{k \notin \Delta} p_k$ where $\Delta \subseteq \{1, 2, ..., |x|\}$, none of $p_j (j \in \Delta)$ holds on $x$, and all $p_k (k \notin \Delta)$ has the form of $f_k = x[k]$. Moreover, if an input $x'$ satisfies a counterfactual explanation $g(x, f)$, then $|f(x) - f(x')| \geq d$ for a given $d$. In the case of binary classification, d is 1.

We define $\mathbb{I}$ as the set of all input data. A perturbation model $per : \mathbb{I} \to 2^{\mathbb{I}}$ describes a set of inputs similar to $x$. A local explanation technique $t$ is parameterized by a perturbation model, and generates a local explanation to a model given an input: $t_{per}(f, x) := g(x, f)$. The local explanation reflects the target model's behaviors over the input space given by the perturbation model. Existing local techniques implement the perturbation models as changing feature values, i.e., $per(x) \subseteq \mathbb{R}^{|x|}$.

As we can see, both the vocabularies and the perturbation models of existing local explanations are limited to describing inputs that have the same lengths as the original input[1]. This limits their effectiveness on models processing sequential data of different lengths.

## 3  OUR FRAMEWORK

We now describe REX. Our goal is to provide a general approach to incorporate temporal information in explanations without heavily modifying the explanation generation technique. We introduce REX in three steps: 1) we introduce the definition of local explanations with temporal information, 2) we describe how to augment existing techniques to generate such explanations, and 3) we describe the workflow of generating explanations with temporal information.

### 3.1  LOCAL EXPLANATIONS WITH TEMPORAL INFORMATION

Our key observation is that while the form of explanation expression varies, the expressions are all built from the corresponding vocabulary. If we can add predicates that reflect temporal information to the vocabulary, then naturally the explanations provide temporal information.

Our temporal predicates describe the temporal relationship between a set of features that satisfy basic predicates. We limit the number of features in a temporal predicate up to two because 1) in most cases, the temporal relationship between two features suffices to cover a large range of inputs of different lengths, therefore providing useful information, 2) humans are bad at understanding high-dimensional information. We give their definitions below:

**Definition 3.1** (1-D Temporal Predicate). A 1-D temporal predicate takes the form of

$$\exists i \in \mathbb{Z}^+ \ such \ that \ f_i \ op_1 \ c \wedge i \ op_3 \ d \quad \text{where } c \in \mathbb{R}, d \in \mathbb{Z}^+, op_1, op_3 \in \{=, >, <, \geq, \leq\}.$$

**Definition 3.2** (2-D Temporal Predicate). A 2-D temporal predicate takes the form of

$$\exists i, j \in \mathbb{Z}^+ \ such \ that \ f_i \ op_1 \ c_1 \wedge f_j \ op_2 \ c_2 \wedge j - i \ op_3 \ d$$
$$where \ c_1, c_2 \in \mathbb{R}, d \in \mathbb{Z} \setminus \{0\}, op_1, op_2, op_3 \in \{=, >, <, \geq, \leq\}.$$

To illustrate the effects of a single feature's absolute position, we use 1-D temporal predicates. To illustrate the effects of the relative position between two features, we use 2-D temporal predicates. Moreover, we can use 2-D temporal predicates when the presence of two features together is important but their order is not. For example, feature $j$ can come before or after feature $i$ when the distance $d$ is negative and $op_3$ is $\geq$. This means the order between these two features does not matter. Compared to conventional feature predicates, temporal predicates no longer describe properties of features at fixed positions in an input. Instead, it requires positions of features to satisfy given temporal constraints.

We now introduce the definition of local explanations with temporal information:

---

[1]When explaining NLP models, some perturbation models allow replacing a word with an empty string. This enables the explanations to cover inputs of shorter lengths to some extent.

**Definition 3.3** (Explanation with Temporal Information). A local explanation with temporal information is a local explanation whose vocabulary consists of regular feature predicates, 1-D temporal predicates, or 2-D temporal predicates.

**Examples.** Consider Input sentence I in Table 1, a 2-D temporal predicate is

$$\exists i, j \in \mathbb{Z}^+ \ such \ that \ f_i = \text{``never''} \wedge f_j = \text{``fails''} \wedge j - i = 1.$$

The explanation generated by Anchors augmented with REX in Table 1 is a conjunction that only consists of the above 2-D predicate. Consider another sentence "I hate that man but love her" which is judged as positive. When applying Anchors with REX on it, the explanation is

$$\exists i \in \mathbb{Z}^+ \ such \ that \ f_i = \text{``love''} \quad \wedge \quad \exists j \in \mathbb{Z}^+ . f_j = \text{``but''} \wedge j \geq 3.$$

The second conjunction above is a 1-D predicate. The sentence contains both "love" and "hate" but is judged as positive because there is a "but" in the middle. On the other hand, the sentence "but I hate that man loves her" is judged as negative. Our 1-D predicate captures such information.

## 3.2 AUGMENTING GENERATION TECHNIQUES

We aim to provide a general approach to extend existing local model-agnostic techniques with temporal information without heavily modifying their algorithms. Our key idea is to extend the vocabulary and perturbation model of a technique. The existing techniques essentially treat the target model as a black box and generate surrogate models as explanations. These explanations are (1) described using features from the vocabulary and (2) trained from input-output pairs obtained from the perturbation model. By modifying only these two components, we can generate explanations augmented with temporal information, while keeping the core algorithms intact.

**Extending Vocabularies.** To incorporate a predicate into a vocabulary, the predicate must be able to serve as a feature of an input. Therefore, provided there's a method to evaluate the predicate on a given input, this incorporation is a lightweight task. The evaluation is easy for our 1-D and 2-D temporal predicates on any model input. Adding temporal predicates to the vocabulary allows the explanation technique to describe the behavior of the target model over variable-length inputs.

**Extending Perturbation Models.** To cover inputs of different lengths, we modify the perturbation models of existing techniques. Concretely, we add a preprocessor over a given perturbation model. For a given input $x$, the preprocessor does two modifications in sequence to generate more inputs: 1) it can delete certain features from the input, and 2) it can switch the positions of two features. Formally, we define $rf : \mathbb{I} \to 2^{\mathbb{I}}$ which returns inputs that remove arbitrary features from a given input; $sf : \mathbb{I} \times \mathbb{Z} \times \mathbb{Z} \to 2^{\mathbb{I}}$ which returns an input that switches the positions of two features of a given input. The preprocessor is a function $prep : \mathbb{I} \to 2^{\mathbb{I}}$ and it is defined follows:

$$prep(x, S_{max}) := \bigcup_{\substack{\hat{x} \in rf(x) \\ n - m \leq S_{max}}} sf(\hat{x}, m, n).$$

To control the number of generated inputs, we add a parameter $S_{max}$ to limit the switching operation to two features that are at most $S_{max}$ apart. In our experiment, we set $S_{max} = 1$. This corresponds to the aforementioned setting of 2-D temporal predicates where the distance $d \geq -1$. Given a perturbation model $per$, a constant $S_{max}$, the new perturbation model our approach generates is

$$per^t(x, S_{max}) := \bigcup_{\hat{x} \in prep(x, S_{max})} per(\hat{x}).$$

Careful readers may have noticed that our augmented perturbation model can be made more complex. For example, one can completely shuffle the features in the input without being limited by $S_{max}$, or add new features besides deleting or switching features. In practice, we find our definition above a good balance among generality, efficiency, and fidelity.

## 3.3 GENERATING EXPLANATIONS WITH TEMPORAL INFORMATION

With REX, we can generate explanations following the scheme of local model-agnostic black-box explanation techniques. The only difference is the extended explanation techniques now include

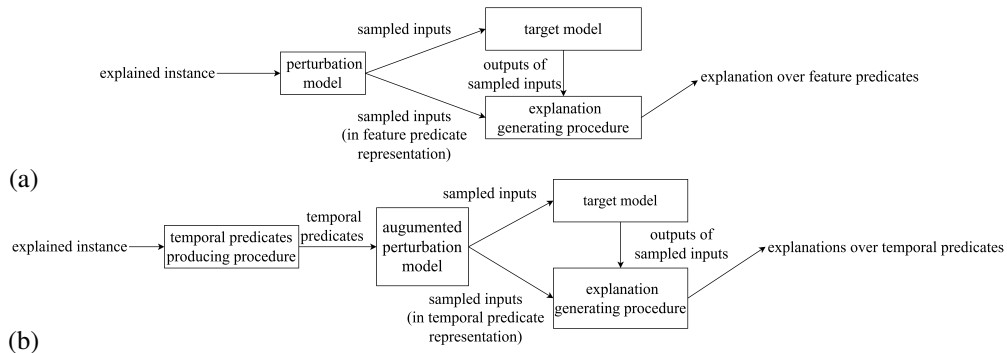

Figure 3: Workflows of generating explanations by local model-agnostic black-box explanation techniques without (a) and with (b) REX.

temporal predicates in their vocabularies. Note the target model still takes the original input. The outputs of the extended perturbation model are also expressed in the original input features, so we can feed them to the target model. Then we evaluate the predicates on these inputs to get the representation in terms of temporal predicates. Figure 3 shows a typical workflow of the explanation-generating process of these techniques without (a) and with (b) REX. After applying REX, the perturbation model provides sampled inputs to the target model to obtain corresponding output, and sampled inputs in the temporal predicate form to generate an explanation with temporal predicates.

## 4 EMPIRICAL EVALUATION

We have instantiated REX on Anchors, LIME, and Kernel SHAP (KSHAP for short) in this section. We refer to the extended version as Anchors*, LIME*, and KSHAP*. We study how much REX improves Anchors, LIME, and KSHAP by comparing them with the extended versions. Following previous works, we use fidelity and understandability (Markus et al., 2021; Alangari et al., 2023) as metrics. They mean good explanations should 1) accurately describe the ML model and 2) be understandable to a human. We conducted two experiments to assess them.

### 4.1 MODELS AND DATASETS

**Sentiment Analysis Models** We trained an LSTM model with paraphrastic sentence embedding (Wieting et al., 2015), and fine-tuned a GPT-2 (Radford et al., 2019) and a Flan-T5 (Chung et al., 2022) model, along with a pre-trained BERT model (HF Canonical Model Maintainers, 2022). We used the Stanford Sentiment Treebank dataset (Socher et al., 2013) and followed the train/validation/test split used in the original dataset. The explanations produced by the original models used only feature predicates, while REX added 1-D and 2-D temporal predicates to the explanations. In the perturbation model, besides deleting and switching words, we also applied BERT (Devlin et al., 2018) to replace words with other words that can appear in the context.

**Anomaly Detection RNN** We trained an Anomaly Detection RNN (Park, 2018) on an ECG dataset (Dau et al., 2018). We again followed the train/validate/test split in the original dataset. We used whether a data point in a series is fixed to a given value as the basic building block of explanations. While the explanations generated by the original techniques only contain such predicates, REX adds temporal predicates. For traceability, we limited the explanations to only consist of data points that are at most 20 steps before the detected anomalous point in a time series. For perturbations, we allowed deleting, switching, and changing data points by sampling from a Gaussian distribution with its original value as the mean and 1 as the standard deviation.

### 4.2 FIDELITY

Fidelity reflects how well an explanation describes the target model. As Anchors provides rule-based explanations while LIME and SHAP provide attribution-based ones, we employed different metrics.

Table 2: Fidelity results.

| Method | Sentiment Analysis | | | | Anom. | Sentiment Analysis | | | | Anom. |
|---|---|---|---|---|---|---|---|---|---|---|
| | LSTM | BERT | T5 | GPT-2 | RNN | LSTM | BERT | T5 | GPT-2 | RNN |
| | Precision (%) | | | | | Coverage (%) | | | | |
| Anchors | 93.6 | 91.1 | **97.9** | 98.6 | **99.3** | 5.1 | 5.7 | 7.6 | 5.9 | 4.6 |
| Anchors* | **95.8** | **94.0** | 97.9 | **99.8** | 93.5 | **12.3** | **14.2** | **10.0** | **10.4** | **8.7** |
| | Accuracy(%) | | | | | AUROC | | | | |
| LIME | 28.8 | 57.2 | 33.6 | 33.9 | 62.3 | 0.505 | 0.519 | 0.505 | 0.513 | 0.575 |
| LIME* | **65.7** | **86.0** | **58.5** | **64.2** | **80.1** | **0.713** | **0.794** | **0.646** | **0.648** | **0.763** |
| KSHAP | 39.8 | 58.9 | 37.0 | 28.1 | 62.9 | 0.493 | 0.514 | 0.521 | 0.577 | 0.557 |
| KSHAP* | **64.2** | **64.1** | **63.0** | **61.1** | **77.4** | **0.603** | **0.682** | **0.652** | **0.687** | **0.716** |

Table 3: Excecution time(seconds) of experiments. "*" indicates the version with REX.

| | Sentiment Analysis | | | | | | | | Anomaly Detection | |
|---|---|---|---|---|---|---|---|---|---|---|
| | LSTM | | BERT | | T5 | | GPT-2 | | RNN | |
| | | * | | * | | * | | * | | * |
| Anchor | **48.3** | 77.2 | **63.1** | 86.0 | **45.2** | 87.2 | **41.3** | 64.2 | 665.3 | **476.4** |
| LIME | **24.9** | 42.5 | **205.5** | 248.3 | **6.8** | 10.7 | **12.9** | 13.2 | **605.2** | 824.2 |
| KSHAP | **10.6** | 12.4 | **88.4** | 139.4 | **12.1** | 23.3 | **187.3** | 189.9 | **630.7** | 869.1 |

For Anchors, we considered the metrics used in rule-based explanation (Lakkaraju et al., 2016; Ribeiro et al., 2018; Craven & Shavlik, 1995) and the sufficient condition nature of Anchors explanations, using **coverage** and **precision** as the fidelity metrics. Given a neural network model $f$, an input instance $x$, and a distribution $D$ corresponding to a perturbation model $pert$, an explanation $A_x$ where $A_x(z) = 1$ if and only if z satisfies the explanation (a sufficient condition), we can define the coverage as $cov(x; f, A) = \mathbb{E}_{z \sim D(x)}[A(z)]$ and precision as $prec(x; f, A) = \mathbb{E}_{z \sim D(x)}[\mathbf{1}_{f(x)=f(z)}|A(z) = 1]$. Coverage is the proportion of input data in the neighborhood space that **match the explanation**. Precision is the proportion of covered data that has **the same model prediction** as the original input.

For LIME and KSHAP, as attribution-based methods, there are primarily two ways to measure the metrics, which are assessing the effect of each high-attribution feature (Hooker et al., 2019; Yoshikawa & Iwata, 2020) and comparing the output of the black-box model with the surrogate model (Balagopalan et al., 2022; Yeh et al., 2019; Ismail et al., 2021). We chose the latter metric for our measurement. The former one needs to change certain predicates while maintaining the others. It is hard for augmented techniques since we cannot change the value of feature predicates without affecting temporal information. Given a black-box model $f$, an input $x$, an explanation surrogate model $E$, and a performance metric $L$ (e.g., accuracy, Area Under the Receiver Operating Characteristic curve (AUROC), or mean squared error), the (in)fidelity is defined as $\mathbb{E}_{z \sim D(x)}L(f(z), E(z))$. In our evaluation, we used accuracy and AUROC as performance metrics.

We took the test sets of the two datasets, and applied Anchors, LIME, KSHAP, Anchors*, LIME*, and KSHAP* to generate explanations for each input in the sets. Since normal points are much more than anomalous points in the anomaly detection dataset, we only looked at anomalous inputs. There are 2210 inputs in the test set of the sentiment analysis dataset, while there are 9 (anomalous) inputs in that of the anomaly detection dataset. To calculate the fidelity, We sampled 10,000 variable-length inputs from the neighborhood space for each input.

Table 2 summarizes the evaluation results. For Anchors, REX improves the coverage by 98.2% on average while maintaining roughly the same level of precision or slightly improving it compared to the original approaches. This is in line with our assumption that REX can augment existing local techniques to explain more inputs. Limited by space, we report the results of 30 randomly chosen inputs in the BERT experiment and all inputs in the anomaly detection experiments in Figure 4. For LIME and KSHAP, REX improves both Accuracy and AUROC, helping to explain the black-box model better by incorporating temporal information. REX improves the accuracy of LIME and Kernel SHAP by 28.6% and 23.1%, and the AUROC by 32.7% and 28.5% respectively.

We did a 1-tail **paired t-tests** on these 15 setup pairs, only differing in the application of REX. The significance values are all far less than 0.01. This shows with over 99% confidence, REX significantly improves the explanation fidelity.

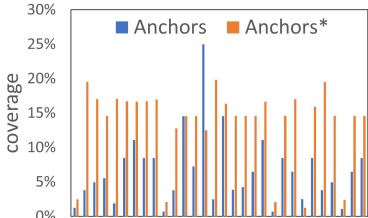 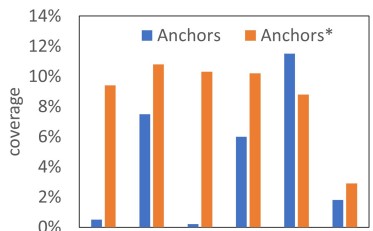

Figure 4: Anchors coverage improvement of 30 inputs in the sentiment analysis BERT experiment and all inputs in the anomaly detection experiment.

Table 4: Results of the user study.

| Method | Precision$_u$ (%) | | | | | Coverage$_u$ (%) | | | | |
|---|---|---|---|---|---|---|---|---|---|---|
| | T1 | T2 | T3 | T4 | T5 | T1 | T2 | T3 | T4 | T5 |
| Anchors | 70.6 | 47.4 | 18.1 | 47.4 | 57.8 | 58.0 | 44.0 | 37.0 | 37.0 | 43.5 |
| Anchors* | 81.2 | 99.4 | 73.9 | 84.1 | 97.7 | 61.9 | 69.5 | 80.5 | 71.5 | 60.5 |

Table 3 reports the execution time. Though REX introduces extra predicates, which potentially lead to increased runtimes, we did not encounter notable issues with efficiency. For LIME and KSHAP, the main time-consuming step is obtaining the results of the target model's output, while REX does not require a larger number of sampled instances, resulting in comparable running times for both techniques. Anchors' runtime heavily depends on how quickly the underlying KL-LUCB algorithm (Kaufmann & Kalyanakrishnan, 2013) can find predicates that fit the model well. Interestingly, when Anchors adds suitable predicates, it can make the algorithm faster. For example, Anchors may not find a good anchor for the sentence "it's not a bad journey at all", but with REX, it can easily and quickly identify the anchor $\{not, bad\} \wedge Pos_{good} - Pos_{bad} \geq 1$ as the explanation. However, if ReX fails to add such predicates, the algorithm may run slower.

### 4.3 UNDERSTANDABILITY

To assess how much REX helps an end user understand a model, we conducted a user study by comparing Anchors* and Anchors. We use $precision_u$ and $coverage_u$ as metrics, similar to those used for measuring fidelity. We employed 19 computer science undergraduates with machine learning backgrounds but no experience with explanation techniques, and studied how much REX improves Anchors on the sentiment analysis LSTM. The questionnaire contains five sets of tests. Each test first presents a sentence, the network's output on the sentence, and explanations from Anchor and Anchor*, respectively. The sentences are randomly chosen from the test set. Then each user is asked to predict the RNN's output on 10 new sentences. The new sentences are produced using our perturbation model (with BERT (Devlin et al., 2018)). They can answer "positive", "negative", or "I don't know". If a user did not answer "I don't know", we gave a "yes" to the $coverage_u$ question; if their prediction matches the actual model output, we gave a "yes" to the $precision_u$ question.

Table 4 shows the average $coverage_u$ and $precision_u$ across the 19 users and 10 sentences for each question. Anchors* is better than Anchors on all questions in terms of both $coverage_u$ and $precision_u$. Across these questions, Anchors* yields an average $precision_u$ of 87.3% and an average $coverage_u$ of 68.8%, while these numbers are only 48.3% and 43.9% for Anchors. The improvements are 80.9% and 56.7% respectively. We did 1-tail **paired t-tests** on these paired data. With more than 99% confidence, REX significantly helps Users predict more instances more precisely.

Users can utilize the explanations to predict the behaviors of the target models on more inputs and more precisely. Incorporating temporal information improves the clarity of explanations (e.g., ones in Table 1 and Figure 1), whereas the new perturbation model considers the fact that a word can appear at any position due to variable input lengths. Strictly speaking, an explanation produced by Anchors only covers inputs with lengths that are equal to the original input's. We inspected the answers closely and found out that sometimes users would misuse the original Anchors' explanations. Some users would ignore input length constraints and apply the explanations when they were supposed to answer "I don't know". This situation happened for 24.8% of all the answers. Therefore, they often give

incorrect predictions to these sentences while the coverage is increased. But such misuses rarely happen with Anchors* because the corresponding explanations highlight temporal information and cover more inputs.

Our user study shows that REX can help users predict more instances with higher $precision_u$. It also shows the shortcomings of existing techniques when applied to models processing sequential data of different lengths. In particular, users are likely to misuse and make wrong predictions when applying the explanations to inputs whose lengths are different from the original input's.

## 5 RELATED WORK

Our work relates to explanation techniques for models processing inputs of variable lengths and general local model-agnostic explanation techniques. We refer to the surveys (Wang et al., 2018; Jacobsson, 2005; Rojat et al., 2021; Molnar, 2020) for comprehensive introductions.

Many works have concentrated on extracting DFAs (Jacobsson, 2005) and their variants (Ayache et al., 2018; Du et al., 2019; Dong et al., 2020) from RNNs or develop more scalable algorithms like adapting Angluin's $L^*$ algorithm (Weiss et al., 2018). These approaches are not practical to complex models due to their global nature and are mostly limited to explaining toy RNNs. Some approaches (Wisdom et al., 2016; Sha & Wang, 2017) prioritize fidelity over understandability by producing surrogate models that require expert analysis. In contrast, REX can help regular end users.

Local explanations for these models mainly attribute importance to features (Arras et al., 2019; Schlegel et al., 2019) like LIME. But they fail to capture temporal information. Treating models as white boxes can lead to more efficient and precise explanations (Li et al., 2016; Denil et al., 2014; Ding et al., 2017; Arras et al., 2017; Arjona-Medina et al., 2019; Murdoch et al., 2018; Karim et al., 2017; Vinayavekhin et al., 2018), but this makes such approaches model-specific. Some consider the effect of multiple features together (Chen et al., 2020; Singh et al., 2018; Sivill & Flach, 2022; Tsang et al., 2020), but these methods ignore temporal information and cannot handle inputs of variable lengths. They extend explanations in an orthogonal direction to REX, so they could potentially be combined with REX to consider both temporal information and multiple-feature effects.

Local model-agnostic explanation techniques apply to a wide range of machine-learning models by treating models as black boxes. Popular explanation forms include feature importance attribution (Ribeiro et al., 2016; Strumbelj & Kononenko, 2014; Lundberg & Lee, 2017; Dandolo et al., 2023), boolean expressions as sufficient conditions (Ribeiro et al., 2018), and counterfactuals (Wachter et al., 2017; Dandl et al., 2020; Zhang et al., 2018). Individual Conditional Expectation plots visualize how the model output changes as one feature changes (Goldstein et al., 2015). These approaches fail to capture temporal information in models processing sequential data of variable lengths.

## 6 CONCLUSION AND LIMITATION

We have proposed REX, a general framework that adds temporal information to existing local model-agnostic explanation methods. REX allows these methods to generate more useful explanations for models that handle inputs of variable lengths (e.g., RNNs and transformers). REX achieves this by extending vocabularies of explanations with temporal predicates, and modifying perturbation models so they can generate inputs of different lengths. We have instantiated REX on Anchors, LIME, and Kernel SHAP, and demonstrated the effectiveness by empirical evaluation.

One limitation of our approach is that it relies on finding realistic perturbation models which is a common problem for model-agnostic explanation techniques. For example, it is unclear what a good perturbation model for the time series data of stock prices is.

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

# A APPENDIX

## A.1 INSTANCES

In this section, we present the instantiation of our framework on several popular model-agnostic local explanation methods (Molnar, 2020). We provide details about the augmented explanations, the integration of temporal predicates, and the interaction between the perturbation model and the core algorithm. Since model-agnostic techniques treat target models as black boxes, they typically sample different inputs and observe how the model outputs change. In our experiment, we implement the instances corresponding to three representative approaches, LIME, kernel SHAP, and Anchors.

**LIME (Ribeiro et al., 2016).** As the example in Figure 1 shows, the augmented explanation is now a linear expression whose 0-1 variables can either be a conventional predicate or a temporal predicate. There are two main steps in LIME to generate explanations: 1) obtaining several sample inputs around a given input using a perturbation model and the corresponding outputs, and 2) performing regression on the input-output pairs. To instantiate REX on LIME, it only needs to be able to evaluate the linear expression and calculate the loss value after adding temporal predicates. For regression algorithms, it is easy to cope with additional predicates. As for the perturbation model, it is used to draw the aforementioned inputs and is viewed as a black box by the core algorithm in LIME. As a result, replacing the perturbation model is straightforward.

**Kernel SHAP (Lundberg & Lee, 2017).** Similar to LIME, kernel SHAP attributes importance to individual features with a regression algorithm, albeit a different loss function, weighting kernel, and regularization term to align the generated explanations with the properties of Shapley values. Consequently, applying our framework to Kernel SHAP resembles the process used for LIME.

**Anchors(Ribeiro et al., 2018).** As Table 1 shows, the augmented explanation is now a conjunction of conventional and temporal predicates. Similar to LIME, Anchors are generated by a sampling-based approach. But in contrast to LIME, Anchors uses if-then rules as its explanation form. Except for the perturbation model, the other parts of Anchors can work well with any predicate. To incorporate temporal information, the new perturbation model only needs to be modified as LIME does. Therefore, adding the temporal predicates and substituting the perturbation model are straightforward.

Besides these methods, other model-agnostic local explanation methods can cooperate with REX in similar ways. Such as Counterfactual Explanations(Wachter et al., 2017; Dandl et al., 2020; Van Looveren & Klaise, 2021), also follow the above algorithm structure. We only need to follow suit while applying REX on these methods.

## A.2 DETAILS ABOUT USER STUDY

The questionnaires are similar for all users with minor variations in the order of presentation. We presented the five questions, Q1, Q2,..., and Q5 in a random order. For each question, we presented the explanations generated without/with ReX in a random order. Also, we presented the ten sentences to be predicted in a random order. We presented one explanation at a time and asked participants to simulate the model on 10 sentences. Since there were 10 explanations in total, we repeated this process 10 times. Figure 5 is a question example in our user study.

**Original sentence**:   pretentious editing ruins a potentially terrific flick.
**RNN output:**   negative.
**Explanation 1**:   the word "ruins" appears in the sentence at the specific position.

Please predict the RNN output on each sentence below according to each explanation. You can answer 0. negative, 1. positive, or 2. I don't know.

| Sentence | Prediction 1 |
|---|---|
| ruins beneath terrific design. | |
| pretentious editing ruins a potentially terrific methodology. | |
| cult ruins a potentially lucrative planet. | |
| ... | |

**Explanation 2**:   both "ruins" and "terrific" appear in the sentence, "terrific" is behind "ruins", and there are at least "0" words between them.

| Sentence | Prediction 2 |
|---|---|
| ruins beneath terrific design. | |
| pretentious editing ruins a potentially terrific methodology. | |
| cult ruins a potentially lucrative planet. | |
| ... | |

Figure 5: A question in the user study.

