# OpenReview forum: "ReX: A Framework for Incorporating Temporal Information in Model-Agnostic Local Explanation Techniques"
_ICLR.cc/2024/Conference — Submitted to ICLR 2024_

### Official Review · Reviewer_9eyb · 2023-10-31

**Soundness:** 2 fair
**Presentation:** 2 fair
**Contribution:** 3 good
**Rating:** 5
**Confidence:** 4

**Summary:**

The paper proposed a general framework for adapting various explanation techniques to models that process variable-length inputs, expanding explanation coverage to data points of different lengths to address limitation of existing model-agnostic general explanation techniques do not consider the variable lengths of input data points, which limits their effectiveness.

**Strengths:**

1)	This paper proposed to incorporate temporal information in local explanations to machine learning models that can capture temporal information in the inputs, which makes these explanations more faithful and easier to understand.
2)	This paper proposed a general framework REX to automatically incorporate the above information in popular local explanation techniques.

**Weaknesses:**

1)	The experimental part is not very sufficient, and the comparison models used are not many and not the most advanced.
2)	After adding the REX framework, the efficiency of the model has decreased significantly, and I feel that it is not very practical.

**Questions:**

1)	After adding the REX framework, the efficiency of the model has decreased significantly. Does this have practical application value?
2)	The reference mentioned in the related work seems to be several years ago. Is there any related work in recent years?
3)	Which models correspond to the methods in Table 4?

---

> ### Author Response · Authors · 2023-11-18
> **Rebuttal by Authors**
>
> Thanks a lot for your review. We address your concerns/questions below:
>
> **[W1]**
>
> The experimental part is not very sufficient, and the comparison models used are not many and not the most advanced.
>
> **[AW1]**
>
> We are conducting an experiment on *AcME—Accelerated model-agnostic explanations: Fast whitening of the machine-learning black box* and will report the results later.
>
> **[Q1]**
>
> After adding the REX framework, the efficiency of the model has decreased significantly. Does this have practical application value?
>
> **[AQ1]**
>
> As we discussed in section 4.2, For LIME and KSHAP, ReX is not the main time-consuming part and does not need any more sampling to get explanations with more fidelity.
>
> For Anchors, the running time heavily depends if there are suitable predicates in the vocabulary. If ReX adds suitable predicates, the Anchors will run faster and otherwise slower.
>
> And as we reported in Table 3, we didn't encounter notable issues with efficiency in experiments.
>
> **[Q2]**
>
> The reference mentioned in the related work seems to be several years ago. Is there any related work in recent years?
>
> **[AQ2]**
>
> To our knowledge, LIME, SHAP, and Anchors are three basic local model-agnostic explanation techniques, and many recent methods are mainly derived from these three methods, such as *TimeSHAP: Explaining Recurrent Models through Sequence Perturbations*, *LIMESegment: LIMESegment: Meaningful, Realistic Time Series Explanations*, and so on. We included the related ones like *LIMESegment* in the related work section.
>
> And we just found a recently released method AcME, we are willing to add it to the related work section.
>
> **[Q3]**
>
> Which models correspond to the methods in Table 4?
>
> **[AQ3]**
>
> As we mentioned in Section 4.3, Table 4 is related to explaining the sentiment analysis LSTM with Anchors and Anchors. We are happy to also add such information in the caption of Table 4.

---

> > ### Author Response · Authors · 2023-11-23
> > **Experimental results**
> >
> > We have finished the experiments on AcME. We use AcME without/with ReX to explain the Bert model on the SST dataset. The results are shown below:
> >
> > | Methods            | AcME | ReX+AcME|
> > | --------         | -------- | -------- |
> > | Accuracy(%)        | 34.7     |  56.4    |
> > | AUROC             | 0.517      | 0.551    |
> > | Time(s)      | 0.45      |  1.87    |
> >
> > ReX can also improve the fidelity of AcME.

---

### Official Review · Reviewer_s4Qe · 2023-11-01

**Soundness:** 2 fair
**Presentation:** 2 fair
**Contribution:** 2 fair
**Rating:** 5
**Confidence:** 4

**Summary:**

This paper introduces a framework, REX, designed to offer model-agnostic explanation techniques for variable-length input data. The framework utilizes both 1-D and 2-D predicates to address the relative positions of individual features within the input data. REX can be plugged into other explanations directly. Specifically, REX is shown to extend the capabilities of Anchors, LIME, and Kernel-SHAP. The efficacy of the proposed method is evaluated on two datasets using metrics including coverage/precision, accuracy, and AUROC. Notably, coverage/precision is also employed as a measure of human interpretability. The results affirm that REX not only enhances the fidelity of explanations but also improves human understanding without imposing additional computational overhead.

**Strengths:**

1. This paper addresses an important research question concerning the impact of temporal relationships between features in the input on the quality of explanations for model decisions. The proposed explanation method is versatile enough to handle varying input lengths, an advantage over traditional model-agnostic explanation techniques. The examples provided offer compelling motivation for the algorithm's design.

2. The experimental evaluation of REX is thorough, covering two distinct tasks: sentiment analysis using multiple language models, and anomaly detection using RNNs. The results are analyzed through a variety of metrics, including both automatic measures of fidelity and human-centric metrics for understanding. The performance improvements achieved by REX are promising.

**Weaknesses:**

1. Algorithm for "Extending Vocabularies": The paper would benefit from a more detailed explanation of how 1-D and 2-D predicates are extracted from the model inputs, as this is a core component of REX. Additional analyses on the algorithm for extending vocabularies would be beneficial. Specifically, do all 1-D and 2-D predicates positively impact explanations? Including more examples beyond those discussed in Section 3.1 could strengthen the paper's argument for the advantages of REX. Additionally, incorporating another textual dataset could bolster claims regarding REX's generalizability.

2. Details on Human Evaluations: The section on human evaluations lacks some details. For instance, is the experimental design a within-groups setup? If so, biases could arise from the order in which explanations from different methods are presented. What is the precise procedure for the user study? Does each set consist of ten new sentences used specifically for that test block? Furthermore, exploring challenging tasks, such as instances where the model makes incorrect predictions, would offer additional insights into human understanding of model reasoning. If humans can also make “wrong predictions” made by the model, it is very convincing that users understand the model's reasoning.

3. Broader Impact of REX: Could REX be generalized to image inputs, particularly simple image sequences with important temporal information for classification? A discussion on the generalizability of REX’s two core modules—Extending Vocabularies and Perturbation Models—could enhance the paper.

**Questions:**

1. Analysis of Table 4: The interpretation of Table 4 in Section 4.3 lacks precision. Specifically, the claimed improvements in precision and coverage, cited as "80.9% and 56.7%" in the text, cannot be observed from the results in Table 4.

2. Question on Input Length: What is the average length of the textual input data? If the text input data consists of long paragraphs, would that impose computational burdens on REX?

---

> ### Author Response · Authors · 2023-11-18
> **Rebuttal by Authors**
>
> Thanks a lot for your review. We address your concerns/questions below:
>
> **[W1]**
>
> Algorithm for "Extending Vocabularies": Additional analyses on the algorithm for extending vocabularies would be beneficial. Specifically, do all 1-D and 2-D predicates positively impact explanations? Including more examples beyond those discussed in Section 3.1 could strengthen the paper's argument for the advantages of REX. Additionally, incorporating another textual dataset could bolster claims regarding REX's generalizability.
>
> **[AW1]**
>
> 1. Do all 1-D and 2-D predicates positively impact explanations?
>
> We agree that there could be some redundant 1-D and 2-D predicates.
>
> But the essential point is that there exists some effective 1-D and 2-D predicates helping to construct explanations with higher fidelity.
> For rule-based methods like Anchors, if a predicate is not contributing to the explanation, it will simply be excluded. Moreover, for an instance that does not need 1-D and 2-D predicates, ReX generates explanation with the original feature predicates.
> For attribution-based like LIME and KSHAP,
>
> This ensures that **ReX does not make an explanation worse**.
>
> We appreciate your suggestion and will expand on this discussion.
>
> 2.
>
> We are conducting experiments on the Llama-2 model and the financial phrasebank datasets. We will report the results later.
>
> **[W2]**
>
> Details on Human Evaluations: The section on human evaluations lacks some details. For instance, is the experimental design a within-groups setup? If so, biases could arise from the order in which explanations from different methods are presented. What is the precise procedure for the user study? Does each set consist of ten new sentences used specifically for that test block? Furthermore, exploring challenging tasks, such as instances where the model makes incorrect predictions, would offer additional insights into human understanding of model reasoning. If humans can also make “wrong predictions” made by the model, it is very convincing that users understand the model's reasoning.
>
> **[AW2]**
>
> We are happy to provide more details.
>
> **The questionnaires are similar for all users with minor variations in the order of presentation.**. We presented the five questions, Q1, Q2,..., and Q5 in a random order. For each question, we presented the explanations generated without/with ReX in a random order. Also, we presented the ten sentences to be predicted in a random order.
>
> Among all the 50 sentences to be predicted, on 19 sentences,  the prediction by LSTM model differs from groundtruth(the sentiment classified by humans). On these sentences ReX can still help people to predict better. Users make the same prediction as the LSTM model on 48.9% of these instances with vallina explanation methods, while ReX improves this to 66.7%.
> In other words, ReX indeed helps humans make “wrong predictions” made by the model.

---

> > ### Author Response · Authors · 2023-11-18
> > **Rebuttal by Authors (Supplementary)**
> >
> > **[W3]**
> >
> > Broader Impact of REX: Could REX be generalized to image inputs, particularly simple image sequences with important temporal information for classification? A discussion on the generalizability of REX’s two core modules—Extending Vocabularies and Perturbation Models—could enhance the paper.
> >
> > **[AW3]**:
> >
> > Thanks for your suggestion. We would add such a discussion in our revised version. Briefly:
> >
> > **ReX can be generalized to image sequence inputs.**
> >
> > We use video classification models as an example.
> >
> > To the best of our knowledge, there are few works especially no model-agnostic techniques focusing on explaining these models. One possible way to apply existing local model-agnostic methods is to consider each frame as a feature and use the presence of these frames as feature predicates in the explanation vocabularies. Formally, for an input $x$, as mentioned in Section 2, feature predicates $\{p_j\}_{j=1,2,...,|x|}$ are defined as $p_j := f_j\ op\ c_j$ where $f_j$ represents the j-th frame and $c_j$ is usually the original value of $f_j$. For perturbation models, existing local techniques change the feature values, i.e. change the image of some frames.
> >
> > To incorporate temporal information, ReX extends the vocabularies and perturbation models in the following way.
> > 1. Extending vocabularies:
> >
> > In Section 3.1, we define 1-D predicates as
> > $$\exists i \in \mathbb{Z}^{+}\ such\ that\ f_i\ op_1\ c \wedge i\ op_3\ d
> >      \quad\text{ where }c \in \mathbb{R}, d\in \mathbb{Z}^{+}, op_1,op_3 \in \{=, >, <,\geq,\leq\}$$
> > and 2-D predicates as
> > $$
> > \begin{array}{r@{}l}
> >     &\exists i, j \in \mathbb{Z}^{+}\ such\ that\ f_i\ op_1\ c_1 \wedge f_j\ op_2\ c_2 \wedge j - i\ op_3\ d \\
> >     &\text{ where }c_1, c_2 \in \mathbb{R}, d\in \mathbb{Z}\setminus\{0\}, op_1, op_2, op_3 \in \{=, >, <,\geq,\leq\}.
> > \end{array}
> > $$
> > We illustrate the effects of a single feature(i.e. frame)'s absolute position by 1-D temporal predicates and the effects of the relative position between two features by 2-D temporal predicates.
> > And we incorporate these predicates into the vocabularies.
> >
> > 2. Extending perturbation models:
> >
> > We can still follow the description in section 3.2, using function $rf: \mathbb{I}\rightarrow 2^{\mathbb{I}}$ to remove features and $sf:\mathbb{I}\times\mathbb{Z}\times\mathbb{Z}\rightarrow 2^{\mathbb{I}}$ to switch features, and then construct a extended perturbation model in the same way.
> >
> > **[Q1]**
> >
> > Analysis of Table 4: The interpretation of Table 4 in Section 4.3 lacks precision. Specifically, the claimed improvements in precision and coverage, cited as "80.9% and 56.7%" in the text, cannot be observed from the results in Table 4.
> >
> > **[AQ1]**
> >
> > As we mentioned in Section 4.3, Table 4 demonstrates that the average $precision_u$ of Anchors* is 87.3% while that of Anchors is 48.3%, so we claim that ReX improves $precision_u$ by 80.9%(**$48.3\times 180.9\\% = 87.3$**). Similarly, the average $coverage_u$ of Anchors* is 68.8\% while that of Anchors is 43.9%, so we claim that ReX improves $coverage_u$ by 68.8%($43.9*156.7\\% = 68.8$).
> >
> > **[Q2]**
> >
> > Question on Input Length: What is the average length of the textual input data? If the text input data consists of long paragraphs, would that impose computational burdens on REX?
> >
> > **[AQ2]**
> >
> > 1. What is the average length of the textual input data?
> >
> > Average length: 19.23
> >
> > 2. If the text input data consists of long paragraphs, would that impose computational burdens on REX?
> >
> > Such long inputs would cause the baseline explanation methods to run longer but REX doesn't make the situation worse.
> >
> > The model-agnostic explanation techniques require to do perturbations on input data. A very long input sentence will increase the time of obtaining the results of the target model's output. And to produce explanations with high fidelity, the number of sampled instances will increase a lot.
> >
> > For ReX, As we discussed in section 4.2, for LIME and KSHAP, ReX is not the main time-consuming part and achieves better fidelity with the same number of samples as the original techniques. For Anchors, the running time heavily depends if there are suitable predicates in the vocabulary. If ReX adds suitable predicates, the Anchors will run faster and otherwise slower.

---

> > > ### Comment · Reviewer_s4Qe · 2023-11-22
> > >
> > > I appreciate the authors' response to my comments. However, there remain a few issues that need further attention. Primarily, my concern regarding the "Extending Vocabularies" algorithm is not addressed and still stands, and additional examples are missing as well (W1). Furthermore, the current response lacks experimental results from an additional dataset, which would significantly enhance the robustness of the study.
> > >
> > > It appears that the authors could utilize more time to refine their work and make it technically sound and solid. Therefore, I would maintain my initial evaluation score

---

> ### Author Response · Authors · 2023-11-23
> **Experimental results**
>
> Thanks for your response.
>
> We have finished the experiments with the following setup:
> 1) explain the Llama-2-7b model with LIME and ReX+LIME on the SST dataset, and
> | Methods            | LIME | ReX+LIME|
> | --------         | -------- | -------- |
> | Accuracy(%)        | 63.5     |  71.8    |
> | AUROC             | 51.7      | 69.0    |
> | Time(s)      | 823.2      |  828.3    |
>
> 2) explain the Bert model with LIME and ReX+LIME on the financial phrasebank dataset.
> | Methods            | LIME | ReX+LIME|
> | --------         | -------- | -------- |
> | Accuracy(%)        | 48.8     |  62.4    |
> | AUROC             | 0.500      | 0.633    |
> | Time(s)      | 486.1 |  480.3    |
>
> On both setups, ReX improves the fidelity of the based methods.

---

### Official Review · Reviewer_zbfs · 2023-11-01

**Soundness:** 3 good
**Presentation:** 3 good
**Contribution:** 3 good
**Rating:** 6
**Confidence:** 3

**Summary:**

This work extends existing explanation methods (Anchors, lime, Kernel Shap) to augment temporal information in the explanations. Temporal information is the explanations is defined with upto 2 features by highlighting the position/distance between features (e.g. position of feature K - position of feature L >= 3). The explanations are evaluated on text and time series data (which is the target domain of such explanations) which show improvements in metrics as well as a user study involving 19 individuals. In the context of explainability, temporal information has earlier been used in timeseries data related to shaplets, albeit in a different manner.

**Strengths:**

Overall, the idea of incorporating temporal information using position/distance between features is simple and novel. Temporal information is useful in general and it does help in minimizing the ambiguity in existing explanations as shown by authors for text and time series examples. The fact that existing explanation methods can be extended simply by using an alternate perturbation approach as claimed by authors, is useful.

**Weaknesses:**

- Empirical evaluation can include more models and datasets.
- The user study could be extended in size and diversity of users. Although positional/distance between features is useful, in case of timeseries the shape of a curve (e.g. shaplets) provides richer information and more useful information to SMEs. It might be good to present use cases where SMEs value the distance information in specific domains.

**Questions:**

- In case of LIME-Text explainer (equivalently for SHAP), words of a sentence are randomly deleted to obtain different binary vectors (1= word present, 0=absent) in order to compute distance between original & perturbed samples and fit a linear model. If the words are shifted, then the binary vectors remain unchanged. It was unclear from text how temporal information is recovered in this setting using the existing ridge regression model used by LIME?

- Compared to the base case of no temporal information, how do (a) number of perturbed samples and (b) number of parameters in the linear model fit by lime/shap, scale when temporal information is requested from the explainer?

---

> ### Author Response · Authors · 2023-11-18
> **Rebuttal by Authors**
>
> Thanks a lot for your review. We address your concerns/questions below:
>
> **[W1]**
>
> Empirical evaluation can include more models and datasets.results later.
>
> **[AW1]**
>
> We are conducting experiments on the Llama-2 model and the financial phrasebank datasets. We will report the results later.
>
> **[Q1]**
>
> In case of LIME-Text explainer (equivalently for SHAP), words of a sentence are randomly deleted to obtain different binary vectors (1= word present, 0=absent) in order to compute distance between original & perturbed samples and fit a linear model. If the words are shifted, then the binary vectors remain unchanged. It was unclear from text how temporal information is recovered in this setting using the existing ridge regression model used by LIME?
>
> **[AQ1]**
>
> The temporal information is recovered using the temporal predicates.
> After LIME is augmented by ReX, the 1-D and 2-D predicates represent the effect of absolute positions and relative positions respectively. And for the corresponding binary vectors, 1 means the predicates evaluate to true, while 0 means false. If a word is shifted, its absolute position and relative position to other words are also changed. Therefore, **the binary vectors corresponding to the predicates about the word's position are changed**, so the temporal information can be recovered using the existing regression model.
>
> **[Q2]**
>
> Compared to the base case of no temporal information, how do (a) number of perturbed samples and (b) number of parameters in the linear model fit by lime/shap, scale when temporal information is requested from the explainer?
>
> **[AQ2]**
>
> (a) We achieve higher fidelity by using **the same** number of perturbed samples.
>
> (b) The number of parameters equals the number of predicates in the vocabulary. For sentiment analysis, we simply use all the feature, 1-D, and 2-D predicates. For anomaly detection, we limit the temporal predicates in a window of 20 steps.

---

> > ### Comment · Reviewer_zbfs · 2023-11-22
> >
> > Thanks for addressing my questions, I will retain my score.

---

> ### Author Response · Authors · 2023-11-23
> **Expeirment results**
>
> Thanks for your response.
>
> We have finished the experiments with the following setup:
> 1) explain the Llama-2-7b model with LIME and ReX+LIME on the SST dataset, and
> | Methods            | LIME | ReX+LIME|
> | --------         | -------- | -------- |
> | Accuracy(%)        | 63.5     |  71.8    |
> | AUROC             | 51.7      | 69.0    |
> | Time(s)      | 823.2      |  828.3    |
>
> 2) explain the Bert model with LIME and ReX+LIME on the financial phrasebank dataset.
> | Methods            | LIME | ReX+LIME|
> | --------         | -------- | -------- |
> | Accuracy(%)        | 48.8     |  62.4    |
> | AUROC             | 0.500      | 0.633    |
> | Time(s)      | 486.1 |  480.3    |
>
> On both setups, ReX improves the fidelity of the based methods.

---

### Official Review · Reviewer_MXvE · 2023-11-04

**Soundness:** 2 fair
**Presentation:** 1 poor
**Contribution:** 2 fair
**Rating:** 3
**Confidence:** 3

**Summary:**

The authors propose REX, a framework that incorporates "temporal information" in local post-hoc explanations of DNNs that can take varying-length sequence data as input.

Specifically, REX provides explanations over a vocabulary of "feature predicates" that specify temporal relationships between features, e.g. "the token at index i is 'never', the token at index j is 'fails', and 'never' occurs immediately before 'fails' (j - i = 1)".  The authors illustrate via examples to calculate existing post-hoc explanation techniques (such as LIME and Anchors) over this new vocabulary of predicates.  The authors demonstrate the value of their new approach by arguing that it results in improvements in fidelity measures and users' performance on a forward simulation task, compared to "naive" application of explanation techniques like LIME and Anchors.

**Strengths:**

1. The authors' predicate definitions (Def. 3.1 and 3.2) are intuitive to interpret (e.g., the 2D predicates can be used to specify the number of tokens between two particular tokens in a sentence).  The authors illustrate the potential utility of attributing importance to such predicates rather than individual features in Figure 1.
2. The authors' experimental results (Table 2 and Figure 4) demonstrate the potential value of REX. The explanations provided by REX have higher coverage, precision, and are more accurate surrogate models compared to the naive explanations provided over the original feature set.

**Weaknesses:**

* **Weakness #1: Clarity.** My primary critique of this work is that I found the present draft difficult to understand.  The notation used was not sufficiently explained, and I found the authors' descriptions of their methodology and experiments to be severely underspecified.  Unless these details are clarified, I do not believe this draft is ready to be published in its present state. Specifically:
  * Section 2 (Notation). The description that you provided in the second paragraph is confusing, and does not clarify exactly how to interpret the notation.  Specifically, what is $d$: is it the order in which the token appears in the sentence? What is the minus sign notation (what is $Pos_g$, and how does it differ from $Pos_f$)? What do the numbers mean (e.g., why is there a 2 in the statement $Pos_{fails} - Pos_{never} >= 2$?)
  * Section 3. Your notation is under-specified. It may be helpful in this section to clarify what the 'features' of the example inputs you presented are (e.g., what are the 'features' of the sentence input "Bob is not a bad boy")? Is $f_j$ in this case the token that appears at index $j$ in the sentence, or something else? Similarly, do your feature predicates $p_j$ compare the value of token at index $j$ to some threshold (e.g. "the token at timestep $t$ = 'fails'", or something else? (EDIT: After reading Section 3.1, it seems like $f_j$ is the value of the token at index $j$?  Can you clarify?)
  * Section 3.2. "Extending vocabularies: I don't understand what is described in this paragraph. What do you mean by 'serve as a feature of an input', and a 'method to evaluate the predicate on a given input'? Do you mean that you construct a new set of indicators for each original datapoint where you evaluate whether the predicate holds for that datapoint, and then learn each "surrogate model" using the set of original features plus the predicates?
  * Section 3.3. I am confused from the explanation provided about how the perturbation model that you've described is used for each of the individual explanation methods.  It is unclear to me how this perturbation method, evaluates the possible importance of all of the possible predicates (from the many possibilities that exist).  Take LIME as an example. I am confused about how being able to generate new datapoints where features are "switched" (for example, say we switch $f_j = c_j$ and $f_i = c_i$, allows us to assign an "importance score" to, for example, the predicate $f_j = c$ AND $f_i = c_i$ AND $i - j >= 1$.  Can you clarify exactly how you calculate the importance scores for each of the 1D and the 2D temporal predicates using such perturbation methods?
  * Section 4.3. Can you clarify how you assigned participant to explanation conditions in the user study? Did a single participant only see explanations from 1 of {Anchor, Anchor*}?  Did you present a single "test", and then ask them to simulate the model on 10 sentences, several different times (so elicited 50 total predictions), or show all 5 "tests" right away and then collect 10 predictions per user?
  * Section 4.3. Can you provide screenshots of your study interface (e.g., how you presented the Anchors explanations to users)? I wonder if simply highlighting the most important tokens identified by Anchors within the existing sentence, like in Figure 4 of [1] (rather than just showing them the rule) may result in similar performance increases as showing them the Anchors* explanation.
* **Weakness #2: No comparison to popular existing post-hoc methods for sequential data.** In your Related Work and in the paper's Introduction, you compare post-hoc feature attribution methods like LIME to DFA methods that explain RNNs.  However, there have been many other post-hoc explanation method techniques that have been proposed to explain sequence models like transformers (most notably, attention [2]). Can you provide additional motivation for the benefits of your proposed method over other existing popular methods?

[1] https://arxiv.org/pdf/2302.08450.pdf

[2] https://arxiv.org/abs/1908.04626

**Questions:**

See the listed Weaknesses for my high-priority questions.

Some additional comments/suggestions I had that are less relevant to my score are:
* Introduction: "the explanations can still be complex as RNNs often fail to internalize 'the perfect regular expressions' and contain noise". Why does failure to learn the true regular expression imply that the network is difficult to explain?
* Figure 2: I am confused about this example.  My understanding is that the anomaly data point is 428, and both Anchors and Anchors* identify only datapoints before this anomaly point.  Wouldn't a more suitable explanation include both datapoints that come before, and after point 428?  Is the problem set-up here that you must predict if timestep $x_t$ is an anomaly, given only the observations that came before it?
* Section 2, nit: isn't it more appropriate for the local explanation to be a function of $f$, i.e., $g(x, f)$?
* Section 3. "Anchors is a conjunction which must evaluate to true on $x$".  Can you define what a "conjunction" is inline?  What do you mean by "evaluate to true on $x$"?
* Section 3.1. Can you provide intuition for the explanation with the "I hate that man…" example?  Why does it make sense for "j >= 3" to be the explanation? This seems less important than "but" coming before "love", or after "hate"?
* Section 4.1. Can you provide more detail about the anomaly detection ECG dataset? Is there only a single "feature" being measured at different timesteps? What is the prediction task here (is it given the measurements $x_1, …, x_{(T - 1)}$, to predict if $x_T$ is anomalous)?
* Section 4.2. Your paper says that "REX improves the coverage by 98.2%".  Do you mean that $1.982 x = y$, where $x$ is the average coverage across all of the models and explanation methods before REX, and $y$ is the average coverage across all of the models and explanation methods after REX?
* Section 4.2. I don't understand the intuition for REX's runtime. You state, "REX does not require a larger number of sampled instances".  Why is this the case?
* For your "it's not a bad journey at all" example in 4.2, I think your explanation is incorrect (the token "good" doesn't appear in the original sentence).
* Section 4.3. You state that users "would ignore input length constraints" when "misusing [the original] Anchors' explanations".  But unless I am mistaken, the original Anchors did not include predicates over the input's length? Can you clarify what you mean by this?'

---

> ### Author Response · Authors · 2023-11-22
> **Rebuttal by Authors (I)**
>
> Thanks a lot for your review. We address your concerns/questions below:
>
> ### [Weakness 1& Questions]
>
> ### a. Section 2 (Notation):  The description that you provided in the second paragraph is confusing, and does not clarify exactly how to interpret the notation. Specifically, what is $d$: is it the order in which the token appears in the sentence? What is the minus sign notation (what is $Pos_g$, and how does it differ from $Pos_f$)? What do the numbers mean (e.g., why is there a 2 in the statement $Pos_{fails}-Pos_{never}>=2$?)
>
> **Answer:**
>
> Sorry for the confusing notation. We would appreciate it if you could point out the specific notations in **the second paragraph of Section 2**. Note the points you raised (e.g., ones about $Pos$) appear in Section 1.
>
> Regarding the notation $Pos_{fails}-Pos_{never} \ge 2$, we have explained this in **the third to the last paragraph of Page 2**. *$Pos_f$ is an integer and means the position of feature $f$*.
>
> Specifically, $Pos_f$ and $Pos_g$ are both integers, representing the position of feature $f$ and $g$ respectively. The minus sign between $Pos_f$ and $Pos_g$ denotes integer subtraction between these two integers. $d$ is an integer. Therefore, $Pos_f-Pos_g \ge d$ implies that the difference between the two numbers is greater than or equal to $d$.
>
> To provide a concrete example, let's consider the sentiment analysis example presented in Table 1. In this case, the features are words from the input. $Pos_{fails}$ represents the position of the word "fails" in the input, while $Pos_{never}$ represents the position of the word "never". Both $Pos_{fails}$ and $Pos_{never}$ are integers. Thus, $Pos_{fails}-Pos_{never}\ge 2$ means that the result of subtracting $Pos_{never}$ from $Pos_{fails}$ is greater than 2.
>
> ### b.Section 3. Your notation is under-specified. It may be helpful in this section to clarify what the 'features' of the example inputs you presented are (e.g., what are the 'features' of the sentence input "Bob is not a bad boy")? Is $f_j$ in this case the token that appears at index $j$ in the sentence, or something else? Similarly, do your feature predicates $p_j$ compare the value of token at index $j$ to some threshold (e.g. "the token at timestep $t$ = 'fails'", or something else? (EDIT: After reading Section 3.1, it seems like $f_j$ is the value of the token at index $j$? Can you clarify?)
>
> **Answer:**
>
> 1. what the 'features' of the example inputs you presented are?
>
> There is a general definition of features that defines them as **attributes of input data.** For example, in image data, features can be pixels, and in sentence data, features can be words. Since this definition of features is widely used and given the limited space, we did not provide a detailed explanation of what features are.
>
> For the example in section 3.1, the "features" refer to the tokens (words) of the example inputs.
>
>
> 2. Is $f_j$ in this case the token that appears at index $j$ in the sentence?
>
> Yes. We have defined $f_j$ in **the last paragraph of page 3**, where we stated that *$f_j$ represents the $j$th feature of input $x$*. In the example, $f_j$ represents the j-th token of the example inputs.
>
>
> 3. do your feature predicates $p_j$ compare the value of token at index $j$ to some threshold.
>
> Yes. We have also defined feature predicates $p_j$ in **the last paragraph** of page 3, where we stated that *$p_j := f_j\ op\ c$*. Here, $p_j$ compares the value of the token at index $j$ to a threshold value denoted as $c$.
>
>
> We are happy to incorporate additional explanations to provide further clarity in the paper.
>
>
> ### c.Section 3.2. "Extending vocabularies: I don't understand what is described in this paragraph. What do you mean by 'serve as a feature of an input', and a 'method to evaluate the predicate on a given input'? Do you mean that you construct a new set of indicators for each original datapoint where you evaluate whether the predicate holds for that datapoint, and then learn each "surrogate model" using the set of original features plus the predicates?
>
> **Answer:**
>
> We define the *vocabulary* in the first paragraph of page 4. The vocabularies are a set of predicates used to construct the explanations.
>
> The original vocabularies only contain **feature predicates**. ReX **extends** the vocabularies by incorporating additional predicates such as the 1-D and 2-D temporal predicates mentioned in **Section 3.1**, so that can incorporate temporal information in the explanation(**Definition 3.3**).
>
> Since the vallina explanation methods use only the original features as predicates, we said we should make new predicates also "serve as a feature".
>
> When utilizing the predicates in the explanation techniques, existing explanation methods need to know if the predicates are true or false, so we need to "evaluate the predicate on a given input".

---

> ### Author Response · Authors · 2023-11-22
> **Rebuttal by Authors (II)**
>
> ### d.Section 3.3. I am confused from the explanation provided about how the perturbation model that you've described is used for each of the individual explanation methods. It is unclear to me how this perturbation method, evaluates the possible importance of all of the possible predicates (from the many possibilities that exist). Take LIME as an example. I am confused about how being able to generate new datapoints where features are "switched" (for example, say we switch $f_j = c_j$ and $f_i = c_i$, allows us to assign an "importance score" to, for example, the predicate $f_j = c$ AND $f_i = c_i$ AND $i - j >= 1$. Can you clarify exactly how you calculate the importance scores for each of the 1D and the 2D temporal predicates using such perturbation methods?
>
> **Answer:**
>
> Let's take ReX+LIME(LIME*) and sentiment analysis as an example to clarify how the perturbation methods calculate the importance scores for each predicates.
>
> For an input sequence $x = w_1w_2w_3$, where $w_i$ represents the $i$-th word in the sequence, we can perform a perturbation by switching the 2nd and 3rd words, resulting in $x' = w_1w_3w_2$.
>
> The feature predicates $f_2 = w_2$ and $f_3=w_3$ will evaluate to **false** on $x'$ since the 2nd feature of $x'$  is $w_3$ while the 3rd one is $w_2$.
>
> For 2-D predicates like
> \begin{array}{r@{}l}
>     &\exists i, j \in \mathbb{Z}^{+}\ such\ that\ f_i\ =w_2  \wedge f_j\ =\ w_3 \wedge j - i\ \ge \ 1 \\
> \end{array}
> will evaluate to false. since there exists $f_3 = w_2$ and $f_2 = w_3$ but $2-3<1$.
>
> LIME samples multiple datapoints using the perturbation model. For each sampled datapoint, LIME* calculates the results of all predicates in the explanation vocabularies, and stores the results as a binary vector, along with the corresponding model output for that datapoint.
>
> This process generates a matrix, as shown below:
>
>
> |     | Predicate 1 | Predicate 2 | ... | Predicate m | Model Output |
> | --- | ----------- | ----------- | --- | ----------- | ------------ |
> | data1 | 1 | 1 | ... | 0 | 1 |
> | data2 | 1 | 1 | ... | 1 | 0 |
> | ... | ... | ... | ... | ... | ... |
> | datak | 1 | 0 | ... | 1 | 1 |
>
> LIME uses a regression algorithm to assign importance scores to each predicate based on this matrix.
>
> ### e. Section 4.3. Can you clarify how you assigned participant to explanation conditions in the user study? Did a single participant only see explanations from 1 of {Anchor, Anchor*}? Did you present a single "test", and then ask them to simulate the model on 10 sentences, several different times (so elicited 50 total predictions), or show all 5 "tests" right away and then collect 10 predictions per user?
>
> **Answer**
>
> We are happy to provide more details about the user study.
>
> 1. Can you clarify how you assigned participant to explanation conditions in the user study?
>
> We showed all users the five questions and 2 explanations. The questionnaires are similar for all users with minor variations in the order of presentation. We presented the five questions, Q1, Q2,..., and Q5 in a random order. For each question, we presented the explanations generated without/with ReX in a random order. Also, we presented the ten sentences to be predicted in a random order.
>
> 2. Did a single participant only see explanations from 1 of {Anchor, Anchor*}?
>
> No, each participant saw both explanations.
>
>
> 3. Did you present a single "test", and then ask them to simulate the model on 10 sentences, several different times (so elicited 50 total predictions), or show all 5 "tests" right away and then collect 10 predictions per user?
>
> We presented one explanation at a time and asked participants to simulate the model on 10 sentences for each explanation. Since there were 10 explanations in total, we repeated this process 10 times.

---

> ### Author Response · Authors · 2023-11-22
> **Rebuttal by Authors (III)**
>
> ### f. Section 4.3. Can you provide screenshots of your study interface (e.g., how you presented the Anchors explanations to users)? I wonder if simply highlighting the most important tokens identified by Anchors within the existing sentence, like in Figure 4 of [1] (rather than just showing them the rule) may result in similar performance increases as showing them the Anchors* explanation.
>
> **Answer**
> Sorry that it is not possible to directly upload images on OpenReview.  However, I can provide a textual representation of how we present the Anchors explanations to the users. The format is as follows:
>
> * **Original sentence:**  pretentious editing ruins a potentially terrific flick.
> * **RNN output:** negative.
> * **Explanation 1:** the word "ruins" appears in the sentence at the specific position.
>
>
> Please predict the RNN output on each sentence below according to each explanation. You can answer 0. negative, 1. positive, or 2. I don't know.
>
>
>
> | Sentence | Prediction 1 |
> | -------- | -------- |
> | ruins beneath terrific design.     |      |
> |  pretentious editing ruins a potentially terrific methodology.     |          |
> | ...|
> | cult ruins a potentially lucrative planet.     |      |
>
> * **Explanation 2**: both "ruins" and "terrific"  appear in the sentence, "terrific" is behind "ruins", and there are at least "0" words between them.
>
> | Sentence |  Prediction 2|
> | -------- | -------- |
> | ruins beneath terrific design.     |      |
> |  pretentious editing ruins a potentially terrific methodology.     |
> | ...|
> | cult ruins a potentially lucrative planet.     |      |
>
> Regarding Figure 4 of [1], if altering the presentation of the explanation leads to performance improvements, it is likely to benefit both Anchors and  ReX+Anchors.
>
> ### g.Introduction: "the explanations can still be complex as RNNs often fail to internalize 'the perfect regular expressions' and contain noise". Why does failure to learn the true regular expression imply that the network is difficult to explain?
>
> **Answer**
>
> In this part of the paper, we discuss using DFAs as global surrogates to explain RNNs.
>
> When using RNNs to learn simple regular expressions, they are often unable to represent regular expressions perfectly and may contain noise, so the decision procedure of these RNNs will be complex. These RNNs are hard to explain by DFAs. As the referred paper *Extracting Automata from Recurrent Neural Networks Using Queries and Counterexamples* concluded, for networks with complicated behavior, extraction becomes extremely slow and returns large DFAs. Large DFAs are also difficult to understand by end users.
>
> ### h.Figure 2: I am confused about this example. My understanding is that the anomaly data point is 428, and both Anchors and Anchors* identify only datapoints before this anomaly point. Wouldn't a more suitable explanation include both datapoints that come before, and after point 428? Is the problem set-up here that you must predict if timestep $x_t$ is an anomaly, given only the observations that came before it?
>
> **Answer**
>
> Due to the workflow of RNNs(https://en.wikipedia.org/wiki/Recurrent_neural_network#/media/File:Recurrent_neural_network_unfold.svg), in this task, we give the anomaly detection RNN inputs data in the time order. **After giving the value of $x_1,x_2,...,x_t$, the RNN will predict if $x_t$ is an anomaly immediately**.
>
> ### i.Section 2, nit: isn't it more appropriate for the local explanation to be a function of $f$, i.e., $g(x,f)$?
>
> **Answer**
>
> Thanks for your suggestion, we will make local explanation a function in our revised version.
>
> ### j.Section 3: Can you define what a "conjunction" is inline? What do you mean by "evaluate to true on $x$"?
>
> **Answer**
>
> In Section 3, we use the common definition of conjunction as provided in the link to Wikipedia (https://en.wikipedia.org/wiki/Logical_conjunction). This definition is widely used and accepted.
>
> Regarding the phrase "evaluate to true on $x$," it simply means that $x$ satisfies the condition described by the predicate $p_j$. For a concrete example, please refer to the example provided in the **Answer of d.**.
>
> ### k.Section 3.1: Can you provide intuition for the explanation with the "I hate that man…" example? Why does it make sense for "j >= 3" to be the explanation? This seems less important than "but" coming before "love", or after "hate"?
>
> **Answer**
>
> Just like your example, the two predicates
>
> 1. "but" is before "love", and
> 2. "but" is after "hate"
>
> can also construct an explanation.
>
>
> However, Anchors tends to find the explanation that covers the most data points with precision >= given threshold. In this example, both this explanation and the one in Section 3.1 can reach the precision threshold, but the one in Section 3.1 covers more data points, so Anchors chooses it as an explanation.

---

> ### Author Response · Authors · 2023-11-22
> **Rebuttal by Authors (IV)**
>
> ### l.Section 4.1: Can you provide more detail about the anomaly detection ECG dataset? Is there only a single "feature" being measured at different timesteps? What is the prediction task here (is it given the measurements $x_1, …, x_{(T - 1)}$, to predict if $x_T$ is anomalous)?
>
> **Answer**
>
> For the anomaly detection ECG dataset, it consists of measurements of a single feature (ECG signal) taken at different time steps.
>
> You can refer to the **Answer of h** and the GitHub repo we referred to in the paper(https://github.com/chickenbestlover/RNN-Time-series-Anomaly-Detection).
>
> ### m.Section 4.2: Your paper says that "REX improves the coverage by 98.2%". Do you mean that $1.982x = y$, where $x$ is the average coverage across all of the models and explanation methods before REX, and $y$ is the average coverage across all of the models and explanation methods after REX?
>
> **Answer**
>
> Yes.
>
> ### n.Section 4.2:  For your "it's not a bad journey at all" example in 4.2, I think your explanation is incorrect (the token "good" doesn't appear in the original sentence).
>
> **Answer**
>
> Thanks for your notification.
>
> That's a typo. It should be $\{not, bad\}\wedge pos_{bad}-pos_{not}\ge 1$.
>
> ### o.Section 4.3：  You state that users "would ignore input length constraints" when "misusing [the original] Anchors' explanations". But unless I am mistaken, the original Anchors did not include predicates over the input's length? Can you clarify what you mean by this?'
>
> **Answer**
>
> As a black-box method, Anchors generates explanations based on the model's predictions on the data points sampled by the perturbation model.
>
>
> In Section 2, we explained that existing local techniques implement the perturbation models by changing feature values, represented as $per(x) \subseteq \mathbb{R}^{|x|}$. Anchors generates explanations by considering only the perturbation space. Consequently, the explanations can only explain the behavior of the model within this space. Specifically, the explanations focus on data points of the same length and variations in the values of some features.
>
> When we mentioned that users "would ignore input length constraints", we meant that they might mistakenly apply the explanations beyond the intended scope. We also addressed these input length constraints in the explanations we provided to our users.
>
> ### [W2] No comparison to popular existing post-hoc methods for sequential data. In your Related Work and in the paper's Introduction, you compare post-hoc feature attribution methods like LIME to DFA methods that explain RNNs. However, there have been many other post-hoc explanation method techniques that have been proposed to explain sequence models like transformers (most notably, attention [2]). Can you provide additional motivation for the benefits of your proposed method over other existing popular methods?
>
> **Answer**
>
> There is a minor misconception that attention-based methods are actually ante-hoc.
>
> The benefits of ReX with existing local model-agnostic methods are that ReX can explain any machine learning model without knowing its internal information, while white-box methods like attention-based methods can only explain a specific kind of models.
>
> But we are happy to compare ReX with these methods. We are conducting an experiment over an attention-based method *Enjoy the Salience: Towards Better Transformer-based Faithful Explanations with Word Salience*.
>
> Thanks for your reading.

---

> ### Comment · Reviewer_MXvE · 2023-11-23
> **Response acknowledged**
>
> Thanks to the authors for responding to all of my listed points.  Their responses clarified several questions I had on details that I found to be unclear.  I especially found the authors' written descriptions of their $Pos_f$ notation, and example of how "ReX+LIME(LIME*)" is calculated to be particularly clarifying.  I also found the descriptions of the user study (and example text that was shown to users) to be illustrative. I'd encourage the authors to add in some of these explanations in a future draft of the paper. I also would have appreciated if the reviewers had updated the draft (rather than retrospectively writing clarifications) before the deadline to do so had passed.
>
> After reviewing the larger discussion (and with my new understanding of how the perturbation model is used to assign importance scores to the predicates), I still have a few remaining questions.  One question which is relevant to reviewer s4Qe's comment requesting clarity on whether 2D predicates are "redundant" with 1D predicates, and how the vocabulary is extended in practice. I also am curious about the extent to which different predicates are highly correlated with each other, which violates LIME's assumption that all of the input features are (somewhat) independent. Does this pose problems when assigning importance scores to highly-correlated predicates? For example, the predicate "token i is in front of token j" is always true when the predicate "token i is at least 2 positions in front of token j" is true.
>
> For the above reasons, I do not recommend the present draft for acceptance and retain my original score.

---

> > ### Author Response · Authors · 2023-11-23
> > **Expeirment results**
> >
> > Thanks for your response.
> > We have conducted the experiment over the attention-based method Enjoy the Salience: Towards Better Transformer-based Faithful Explanations with Word Salience, which proposed Saloss. We compared Saloss with ReX+LIME. The results are shown as follows:
> >
> > | Methods            | Saloss | ReX+LIME|
> > | --------         | -------- | -------- |
> > | Accuracy(%)        | 48.8     | 86.0    |
> > | AUROC             | 0.501      | 0.794    |
> > | Time(s)      | 6.59 |  248.3    |
> >
> > As white-box methods, Saloss has better efficiency than ReX, but using only attention attribution cannot reach a high fidelity on sequence data.

---

### Meta-Review · Area_Chair_Jn2s · 2023-12-12

**Metareview:**

This paper enters the fraught area of "local explanation" for which their are many well-known methods (e.g., LIME, SHAP, ANCHORS) but no well-known problems for which they are solutions. Moreover these methods are known to have failed nearly every test of efficacy and to disagree with each other. The paper approaches this literature uncritically, repeating unsubstantiated claims, e.g. that the methods help practitioners to "judge whether the results are trustworthy" and "understand knowledge embedded in the systems so they can use the knowledge to manipulate future event". This latter point is a flagrant inaccurate causal claim. The authors cite others to back up their claims but unfortunately, in an area where the literature is this messy, one cannot hide behind citations. The authors proceed, suggesting to augment these methods with "temporal information". The paper received four reviews, three of which advocate rejection. Overall the reviewers, while less critical than I of the initial framing, found the paper to be unclear, did not feel that the algorithm was described properly, and this opinion was sustained despite an interactive discussion period. On this basis I am recommending rejection.

**Justification For Why Not Higher Score:**

Unclear paper, misleading introduction, not suitable for publication.

**Justification For Why Not Lower Score:**

N/A

---

### Decision · Program_Chairs · 2024-01-16

Reject